# GPT-4 generates accurate and readable patient education materials aligned with current oncological guidelines: A randomized assessment

Severin Rodler[1,2,3☉], Francesco Cei[1,2,4], Conner Ganjavi[1,2☉], Enrico Checcucci[5], Pieter De Backer[6,7], Ines Rivero Belenchon[8], Mark Taratkin[9], Stefano Puliatti[10], Alessandro Veccia[11], Pietro Piazza[12], Loïc Baekelandt[13], Karl-Friedrich Kowalewski[14,15,16], Juan Gómez Rivas[17], Christian D. Fankhauser[18], Marco Moschini[4], Giorgio Gandaglia[4], Riccardo Campi[19], Andre De Castro Abreu[1,2], Giorgio I. Russo[20], Andrea Cocci[21], Serena Maruccia[22], Giovanni E. Cacciamani[1,2*], YAU Collaborators[¶]

1 USC Institute of Urology and Catherine and Joseph Aresty Department of Urology, University of Southern California, Los Angeles, California, United States of America, 2 Artificial Intelligence Center at USC Urology, USC Institute of Urology, University of Southern California, Los Angeles, California, United States of America, 3 Department of Urology, University Hospital Schleswig-Holstein, Campus Kiel, Kiel, Germany, 4 Department of Urology, IRCCS San Raffaele Hospital and Vita-Salute San Raffaele University, Milan, Italy, 5 Department of Surgery, Candiolo Cancer Institute, FPO-IRCCS, Turin, Italy, 6 Department of Urology, Onze-Lieve-Vrouwziekenhuis Hospital, Aalst, Belgium, 7 ORSI Academy, Ghent, Belgium, 8 Urology and Nephrology Department, Virgen del Rocío University Hospital, Seville, Spain, 9 Institute for Urology and Reproductive Health, Sechenov University, Moscow, Russia, 10 Department of Urology, University of Modena and Reggio Emilia, Modena, Italy, 11 Urology Unit, Azienda Ospedaliera Universitaria Integrata Verona, Verona, Italy, 12 Division of Urology, IRCCS Azienda Ospedaliero-Universitaria di Bologna, Bologna, Italy, 13 Department of Urology, University Hospitals Leuven, Leuven, Belgium, 14 Department of Urology, University Medical Center Mannheim, University of Heidelberg, Mannheim, Germany, 15 Mannheim, Institute of Medical Biometry, University of Heidelberg, Heidelberg, Germany, 16 DKFZ Hector Cancer Institute at the University Medical Center Mannheim, Mannheim, Germany, 17 Department of Urology, Hospital Clinico San Carlos, Madrid, Spain, 18 Department of Urology, University of Zurich, Zurich, Switzerland, 19 Unit of Urological Robotic Surgery and Renal Transplantation, Careggi Hospital, University of Florence, Florence, Italy, 20 Urology Section, University of Catania, Catania, Italy, 21 Department of Urology, University of Florence, Careggi Hospital, Florence, Italy, 22 Istituti Clinici Zucchi - Gruppo San Donato, MB, Italy

¶ Full YAU Collaborator list can be found in the acknowledgments.
☉ These authors contributed equally to this work.
* Giovanni.cacciamani@med.usc.edu

## Abstract

### Introduction and aim

Guideline-based patient educational materials (PEMs) empower patients and reduce misinformation, but require frequent updates and must be adapted to the readability level of patients. The aim is to assess whether generative artificial intelligence (GenAI) can provide readable, accurate, and up-to-date PEMs that can be subsequently translated into multiple languages for broad dissemination.

**Data availability statement:** All relevant data are within the manuscript and its Supporting Information files.

**Funding:** The author(s) received no specific funding for this work.

**Competing interests:** The listed authors of this manuscript have the following competing interests: Giovanni Cacciamani holds equity in EditorAI Pro. This does not alter our adherence to PLOS ONE policies on sharing data and materials. There are no patents, products in development or marketed products associated with this research to declare.

## Study design and methods

The European Association of Urology (EAU) guidelines for prostate, bladder, kidney, and testicular cancer were used as the knowledge base for GPT-4 to generate PEMs. Additionally, the PEMs were translated into five commonly spoken languages within the European Union (EU). The study was conducted through a single-blinded, online randomized assessment survey. After an initial pilot assessment of the GenAI-generated PEMs, thirty-two members of the Young Academic Urologists (YAU) groups evaluated the accuracy, completeness, and clarity of the original versus GPT-generated PEMs. The translation assessment involved two native speakers from different YAU groups for each language: Dutch, French, German, Italian, and Spanish. The primary outcomes were readability, accuracy, completeness, faithfulness, and clarity. Readability was measured using Flesch Kincaid Reading Ease (FKRE), Flesch Kincaid Grade Level (FKGL), Gunning Fog (GFS) scores and Smog (SI), Coleman Liau (CLI), Automated Readability (ARI) indexes. Accuracy, completeness, faithfulness, and clarity were rated on a 5-item Likert scale.

## Results

The mean time to create layperson PEMs based on the latest guideline by GPT-4 was 52.1 seconds. The readability scores for the 8 original PEMs were lower than for the 8 GPT-4-generated PEMs (Mean FKRE: 43.5 vs. 70.8; $p < .001$). The required reading education levels were higher for original PEMs compared to GPT-4 generated PEMs (Mean FKGL: 11.6 vs. 6.1; $p < .001$). For all urological localized cancers, the original PEMs were not significantly different from the GPT-4 generated PEMs in accuracy, completeness, and clarity. Similarly, no differences were observed for metastatic cancers. Translations of GPT-generated PEMs were rated as faithful in 77.5% of cases and clear in 67.5% of cases.

## Conclusions and relevance

GPT-4 generated PEMs have better readability levels compared to original PEMs while maintaining similar accuracy, completeness, and clarity. The use of GenAI's information extraction and language capabilities, integrated with human oversight, can significantly reduce the workload and ensure up-to-date and accurate PEMs.

## Patient summary

Some cancer facts made for patients can be hard to read or not in the right words for those with prostate, bladder, kidney, or testicular cancer. This study used AI to quickly make short and easy-to-read content from trusted facts. Doctors checked the AI content and found that they were just as accurate, complete, and clear as the original text made for patients. They also worked well in many languages. This AI tool can assist providers in making it easier for patients to understand their cancer and the best care they can get.

## Introduction

Health literacy remains an often-overlooked factor in health disparities, with research directly linking low literacy to negative health outcomes [1,2]. This is in part due to the dearth of validated medical information that is written specifically for patients, forcing many of them to turn to less reliable sources on the internet and potentially compromise care [3]. Approximately 4.5% of all internet search queries are health-related, totaling around 6.75 million health-related searches each day [4]. Given this, countering the exposure to potential misinformation through the creation of readable, correct, and up-to-date patient educational materials (PEMs) is a fundamental objective of societies across specialties [5].

PEMs are drafted by a panel of experts who extract information from clinical guidelines and translate this information, together with a panel of laypersons and patient advocates, into easy-to-understand PEMs [6]. Given the dynamic nature of uro-oncological guidelines, which are updated on a yearly basis, PEMs must be regularly updated to reflect the latest evidence [7]. Moreover, it has been noted that certain sections of PEMs exhibit poor readability, not meeting the readability standards set by the European Union (EU) for patient facing material [8].

Generative artificial intelligence (GenAI), has emerged as a powerful tool for generating patient information However, GenAI outputs often lack consistent accuracy, particularly when dependent solely on the training data of the foundational model [9]. Interestingly, a key feature of GenAI is extracting data from complex input in the form of tokens and assembling those tokens in a logical way [10]. Further, GenAI has great language capabilities, simplifying complex text and adapting it to different literacy levels, and is particularly suitable for laypersons such as patients and their caregivers [11]. However, a global survey of urologists found that while GenAI is useful for patient education, there are ethical concerns regarding the potential spread of biased information and patient privacy breaches [12].

In this study we assessed the potential of GenAI to extract guideline information from the European Association of Urology (EAU), simplify it for better patient understanding, and translate it into multiple languages – bridging the accessibility gap for certified healthcare content. Previous studies have shown that GenAI tools like ChatGPT can be used to simplify PEMs in multiple specialties with comparable correctness and improved readability [13,14]. While these studies relied on the base GenAI model, our study represents a novel approach through the creation of a custom model to extract information from recent urologic oncology guidelines, mimicking the standard workflow for PEM development. We hypothesize that GenAI-generated PEMs not only offer superior readability but also maintain the same level of quality as those traditionally authored by leading urological societies, with the added capability of quality translation into various languages for wider dissemination.

## Materials and methods

This study comprises a mixed methodology consisting of four distinct components: a) GPT-4 framework development and custom GPT creation, b) survey piloting, c) randomized assessment among board certified physicians and urologists to assess correctness – including accuracy, completeness, and clarity - of PEMs, d) bilingual physicians also assessed the faithfulness and clarity of translations into 5 EU languages. Definitions of these metrics are reported below.

This study follows the Checklist for Reporting Results of Internet E-Surveys (CHERRIES) [15] criteria (S1 Table). According to the guidelines of the local institutional review board (IRB), no ethics approval was required for this study as the study has not involved human data in the analysis. No additional support from the EAU offices or external grants was received for the purpose of this study.

### GenAI framework development

For this study, we developed a tri-phasic Generative AI (GenAI) framework (Fig 1). GPT-4 (OpenAI Inc., San Francisco, CA, USA) was utilized to extract relevant information from clinical guidelines (phase 1) and adapt it to the readability level recommended by the European Union (phase 2), as well as to translate the English output into languages commonly spoken within the EU (phase 3). The prompts used were based on previously published methods [11] and were modified

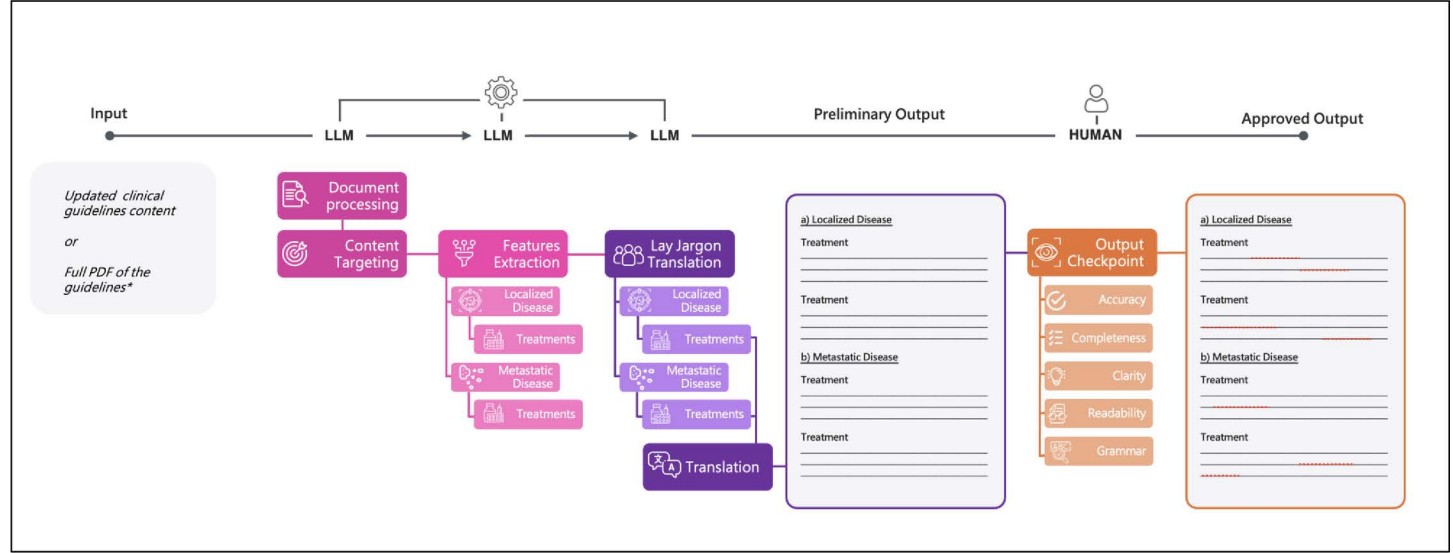

**Fig 1. Pipeline for generating layperson summaries of medical trials using a tri-phasic large language model (LLM)-based framework with human oversight.** The process starts with input from the most current guidelines, which the LLM processes in stages. First, key sections (localized disease, metastatic disease and treatments of each section) are identified (phase 1) and simplified into lay terms (phase 2). The output is then translated into five languages commonly spoken in the EU (phase 3). A human reviewer then evaluates the preliminary output for accuracy, completeness, clarity, and readability. Once approved, the final output is generated, ensuring the summary is both accessible and precise. This method combines GAI efficiency with human quality control to produce reliable patient education materials.* We tested this pipeline's uploading content separately for each disease namely prostate cancer, kidney cancer, bladder cancer, testicular cancer.

to extract key guideline information and generate an overview of therapeutic recommendations for both localized and metastatic disease, written to compile with the EU-recommended patient reading level. The prompts were refined until a consistent output in three consecutive tests was achieved. Temperature and token limits were not adjusted from the base GPT-4 model. A second prompt was created to translate this overview into five EU languages: Dutch, French, German, Italian, and Spanish. At the time of the writing of the present manuscript, the system we created can translate and extract information in 27 languages. The generated outputs were analyzed without further edits, and the time required to produce the output—measured from initiation to completion—was recorded using a digital chronometer (Apple Inc., Cupertino, CA, USA). A custom GPT, utilizing the same prompt for generating PEMs, was developed for free use and to validate the findings of this study: https://chatgpt.com/g/g-sZZBNfGSI-patient-education-material-readible-summaries. The prompt for translating the GenAI-generated PEMs can be found in S2 Table.

### PEMs assessment

**Data sources.** PEMs were retrieved via the official patient information website of the European Association of Urology (https://patients.uroweb.org) on October 10, 2023 by one study investigator (SR) as previously reported [8]. We selected the information for the management of common urologic cancers - prostate cancer, bladder cancer, renal cell carcinoma, and testicular cancer. Only PDF text files were retrieved for PEMs. The guidelines for prostate cancer, bladder cancer, renal cell carcinoma, and testicular cancer were accessed in full-text version from the EAU guideline office based on the most up-to-date version available. At the time of the present study, the EAU guidelines are freely available and downloadable from the official EAU official website (https://uroweb.org/guidelines). One study investigator (SR) accessed the EAU Guidelines website on September 14th, 2023, to retrieve guidelines on prostate cancer, bladder cancer, renal

cancer, and testicular cancer. The generation of GenAI PEMs was completed on October 17th, 2023. None of the retrieved materials were used to train the GenAI pipeline; instead, they were solely utilized to assess its performance. Based on the results of this study, no further iterations were deemed necessary. Only the first output of the GenAI PEMs was used in subsequent stages of the study, and no edits were made to the GenAI-generated PEMs to avoid introducing any bias regarding their correctness trifecta.

This study focuses on the PEMs for the treatment of localized disease and first-line treatment for metastatic disease, as patients most often search the internet for information regarding disease treatment specifically [16–20].

**PEMs readability assessment.**  Readability of PEMs from the EAU patient office and output of GPT-4 was assessed via an automated webtool (www.webfx.com) as previously described [21]. Readability score [Flesch Kincaid Reading Ease (FKRE)] as well as reading education levels [Flesch Kincaid Grade Level (FKGL), Gunning Fog Score (GFS), SMOG Index (SI), Coleman Liau Index (CLI), Automated Readability Index (ARI)] were used for analysis. A high readability score and a low reading education level indicate a good readability.

**PEMs correctness assessment.**  The correctness of the PEMs was assessed independently by each board-certified physician or urologist. Accuracy was defined as the validity of the output information compared to the current guidelines' indications and level of evidence. Completeness was defined as the PEM addressing the currently available treatment options for the given cancer type, compared to the current guidelines' indications and level of evidence. Clarity was defined by the PEM's level of coherently conveying intelligible information to the reader. Accuracy, completeness, and clarity were rated on a 5-item Likert scale (1 = strongly disagree, 2 = disagree, 3 = neither agree nor disagree, 4 = agree, 5 = strongly agree) [22]. A score of ≥ 4 was defined as agreement. The correctness trifecta was defined as agreement in all three metrics. Each physician and urologist independently reviewed, in a randomized fashion, both PEMs generated by GenAI and those drafted by the European Urology Patient Information Committee.

**Pilot assessment of PEMs correctness.**  For the quality assessment of the PEMs, we enrolled members from the Young Academic Urologists working groups. First, 12 members of the Young Academic Urologists (YAU) group for urotechnology and Digital Health (all board-certified physicians or board-certified Urologists) assessed the correctness of GPT-4 generated PEMs compared to original PEMs. Raters received the original and GPT4-generated PEMs in a blinded but not-randomized way in the piloting experiments. Two surveys with 8 PEMs each (original and GPT-4 generated) were distributed. Google forms (Alphabet Inc., Mountain View, CA, USA) was used as the software to collect answers. Aim of the study, study coordination, and definitions of items as described above were provided on the first page of the survey (S1 Table). Only one PEM was depicted per page and ratings for accuracy, completeness and clarity were immediately collected on the same page. Selection of one option per question was enforced. No review step was included in the flow of questions. Unique users were identified by name without the need for cookies, IP addresses, or log file analysis. Only completed questionnaires could be submitted. After collection, all responses were exported from google forms and saved with password protected storage. The respondent's name was stored along with the results to prevent duplicate entries and were deleted prior to analysis. Prior to the start of the study, the survey was tested for the correct display of questions and functionality. No cut-off time was defined for answering the questions.

The survey was sent to recipients via e-mail and closed after all answers were received. No incentives were provided. Data was collected between October 26th, 2023 and November 1st, 2023.

**Randomized assessment of PEMs correctness.**  The Correctness trifecta of the GPT4-generated output was next assessed by 20 members of 4 different indication-specific YAU groups [prostate cancer (n = 5), urothelial carcinoma (n = 5), renal cell carcinoma (n = 5), penile & testis cancer (n = 5)]. Research indicates a lack of consensus on the optimal number of expert evaluators needed, with findings suggesting that 5 reviewers are sufficient to control chance agreement [23,24]. Experts from each YAU working group were selected by the respective YAU chairs as best suited to evaluate PEMs. Our resulting expert panel represents 14%, 19%, 17%, and 22% of prostate, urothelial, kidney, and testis cancer working groups respectively.

 

PEMs were presented in a single-blinded, randomized manner to reduce bias through detection of patterns by respondents. Randomization between original PEMs and GPT-4 generated PEMs was performed by a random number generator in Microsoft Excel (Microsoft, Redmont, WA, US), which determined the placement of each PEM in the survey. Selected members of the YAU groups were provided with the online survey via e-mail as described. The same definitions for correctness metrics were supplied. Google forms (Alphabet Inc., Mountain View, CA, USA) was used and the survey was closed after all answers were received. No incentives were provided. Data was collected between January 15th, 2024, and March 1st, 2024.

**Assessment of GenAI-translated PEMs.** Translations to Dutch, French, German, Italian and Spanish were assessed for faithfulness and clarity on a 5-item Likert scale (1 = strongly disagree, 2 = disagree, 3= neither agree nor disagree, 4 = agree, 5 = strongly agree). Faithfulness was defined as the level of accuracy and completeness compared to the English PEM and clarity was defined as the level of coherence as described above. An expert team approach to language translation was used for independent evaluation with two bilingual members originating from different YAU groups who speak the respective language at a native fluency [25]. The same methodology to conduct the survey was used. The survey link was provided via e-mail to the respective native speakers. No randomization of the PEM was performed, as this part of the analysis only included a direct comparison of GPT-4-generated English PEMs and GPT-4-translated PEM. Data was collected between October 26th, 2023, and March 4th, 2024.

## Statistical analysis

Descriptive statistics were calculated as mean and standard deviation (SD) for interval scaled or as medians for ordinal scaled data. Mann-Whitney-U test was used to analyze differences between original PEMs and GPT-generated PEMs. Chi-Square test was used to compare agreement (4 or 5) versus non-agreement (1–3) between original and GPT-4 generated PEMs. Interrater reliability was calculated by Fleiss' kappa. A p-value equal to or below 0.05 was considered as statistically significant. All analyses were performed using GraphPad Prism version 10.0.0 for Mac (GraphPad Software, Boston, MA, USA) and SPSS Statistics version 26 for Mac (IBM, Armonk, NY, USA).

## Results

A total of 48 PEMs documents were generated, respectively four for localized and four for metastatic diseases in English as well as translations to Dutch, French, German, Italian and Spanish. The mean time to create layperson PEM based on the latest guideline provided by the EAU by the LLM was 52.1 (SD: 6.8) seconds. Original PEMs were retrievable for English and German for all 4 cancer types, while it was available in Spanish for two (50%) and for Dutch, Italian and French in only one cancer type (25%, S3 Table).

### Readability assessment of English PEMs

The readability scores for original PEMs [Mean (SD) Flesch Kincaid Reading Ease: 43.5 (10.8)] were lower than for GPT-4 generated PEMs [Mean (SD) Flesch Kincaid Reading Ease: 70.8 (4.9), p < .001]. In line, the required reading education levels were higher (less readable) for original PEMs [FKGL: 11.6 (2.4); GFS: 14.3 (2.5); SI: 10.6 (1.8); CLI: 14.2 (1.5); ARI: 11.6 (3.0)] compared to GPT4-generated PEMs [FKGL: 6.1 (0.9), p < .001; GFS: 7.4 (0.8), p < .001; SI: 6.3 (0.6), p < .001; CLI: 11.7 (1.3), p = .004; ARI: 6.1 (1.3), p < .001, see Fig 2].

### Correctness assessment of PEMs

As a pilot, 12 urologists of the YAU urotechnology group rated accuracy, completeness, and clarity blinded for the origin of the PEM. In this pilot experiment, the mean (SD) accuracy [original: 3.6 (0.8) vs GPT-4: 3.6 (1.1)], completeness [original: 3.6 (1.1) vs GPT-4: 3.8 (1.1)], and clarity [original: 3.9 (0.8) vs GPT-4: 3.8 (1.0)] for original PEM as well as GPT-4 generated PEM was not significant different for each scenario with an Fleiss' kappa of -0.027 (p = .062, S1 Fig). Baseline characteristics of the rating groups are depicted in S4 Table.

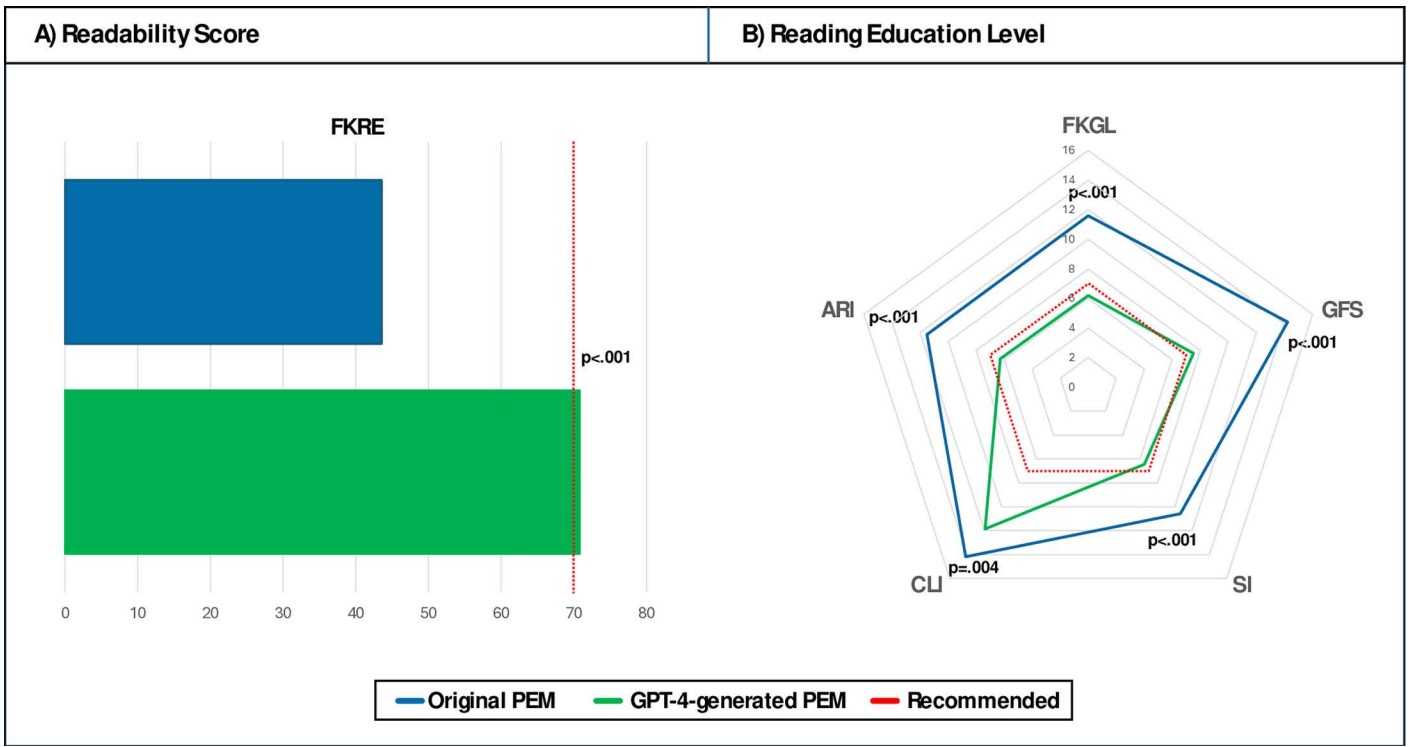

**Fig 2. Readability assessment of patient educational materials (PEM).** Readability scores (A) and reading education levels (B) were determined for original patient education materials (PEMs) and GPT-4 generated PEMs. A high readability score and low reading education level indicate good readability. Recommended readability levels for clinical summaries issued by the European Union are highlighted in red. Abbr.: Patient educational material (PEM), Flesch Kincaid Reading Ease (FKRE), Flesch Kincaid Grade Level (FKGL), Gunning Fog Score (GFS), Smog Index (SI), Coleman Liau Index (CLI), Automated Readability Index (ARI).

Afterwards, twenty urologists with a proven track record in the respective field of uro-oncology and blinded for origin of the PEM rated the accuracy, completeness, and clarity in a randomized way (Fleiss' kappa: 0.007, p = .460). For localized disease, the original PEM was not significantly different from the GPT-4 generated PEM for mean (SD) accuracy [prostate cancer: 4.3 (0.7) vs 3.7 (1.1), p = .122; testicular cancer: 4.0 (0.8) vs 3.7 (1.0), p = .286; renal cell carcinoma: 4.1 (0.9) vs. 4.1 (0.9); p = .960; bladder cancer: 4.1 (0.9) vs 3.6 (0.9), p = .065], completeness [prostate cancer: 4.2 (0.7) vs 3.7 (1.0), p = .126; testicular cancer: 4.1 (0.9) vs 3.6 (0.9), p = .101; renal cell carcinoma: 4.0 (0.8) vs. 4.0 (1.0); p = .982; bladder cancer: 3.8 (1.1) vs 3.8 (0.9), p = .628] and clarity [prostate cancer: 4.4 (0.7) vs 3.7 (1.2), p = .065; testicular cancer: 4.2 (0.9) vs 3.7 (1.2), p = .147; renal cell carcinoma: 4.2 (0.7) vs. 4.0 (1.0); p = .966; bladder cancer: 4.0 (0.9) vs 3.6 (0.9), p = .170]. In line, no differences were observed for metastatic disease for accuracy [prostate cancer: 3.7 (1.1) vs 3.8 (0.9), p = .996; testicular cancer: 3.5 (1.2) vs 3.7 (0.9), p = .704; renal cell carcinoma: 3.5 (1.2) vs. 3.9 (0.9); p = .301; bladder cancer: 3.5 (1.1) vs 3.9 (0.8), p = .205], completeness [prostate cancer: 4.0 (0.9) vs 3.5 (1.0), p = .181; testicular cancer: 3.3 (1.2) vs 3.4 (1.0), p = .935; renal cell carcinoma: 3.8 (1.2) vs. 3.7 (1.0); p = .738; bladder cancer: 3.5 (1.1) vs 3.8 (1.0), p = .451] and clarity [prostate cancer: 4.0 (0.5) vs 3.9 (0.9), p = .900; testicular cancer: 3.5 (1.1) vs 3.4 (1.2), p = .947; renal cell carcinoma: 3.5 (1.2) vs. 3.9 (0.8); p = .803; bladder cancer: 3.7 (0.9) vs 3.9 (1.1), p = .292 see Fig 3). The correctness trifecta was reached in 92/160 rated original PEM and 81/160 rated GPT-4 generated PEM (p = .217).

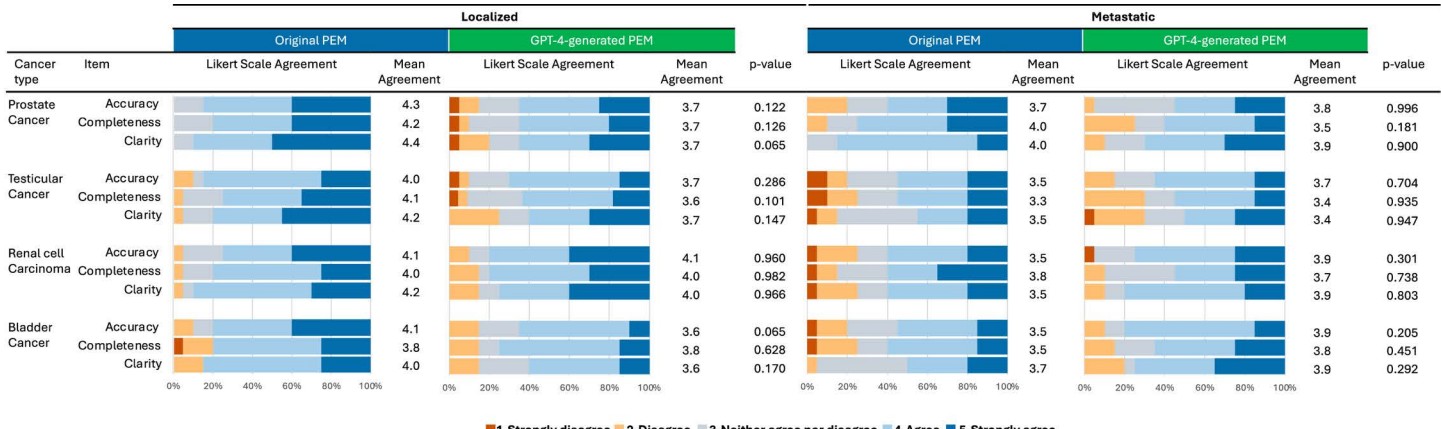

**Fig 3. Correctness assessment of English patient education materials (PEM).** Original patient educational materials (PEM) and GPT-4 generated PEM were used for analysis. Information for both localized and metastatic disease were evaluated for accuracy, completeness, and clarity on a 5-item likert scale. Evaluation was performed in a single-blinded, randomized manner. Abbr.: Patient educational material (PEM).

## Assessment of translation of PEMs by GenAI

Ten urologists rated the faithfulness and clarity of PEM translated from the GPT-4 generated PEMs to Dutch, German, French, Italian and Spanish. Mean (SD) grammar correctness was 3.8 (0.8) and mean (SD) clarity was 3.9 (0.8). Translated PEMs were rated as faithful in 62/80 (77.5%) cases and clear in 54/80 (67.5%) cases for the different languages assessed. Details are reported in Fig 4.

## Discussion

In this study we assessed the performance of a tri-phasic GenAI pipeline in generating precise PEMs within 1 minute that is more readable based on objective measures than conventional PEMs. Accuracy, completeness, and clarity of the information provided is comparable to original PEMs. Translations of GenAI-generated PEMs were rated as clear and faithful in more than two-thirds of cases, enabling the potential dissemination within the member states of the EAU after further refinement. Therefore, integration of GenAI into the generation of PEMs, with current clinical guidelines on treatment provided as the knowledge base, might reduce workload in the creation of this patient-oriented information, while optimizing the availability of high-quality PEMs (Fig 1).

Seeing a physician and discussing treatment options right after being diagnosed with localized or advanced cancer is a key step in a patient's care. For many patients, this may be their first time dealing with the healthcare system, and they are just starting to develop a relationship with their urologist [26]. This critical point in the "cancer journey" is often vulnerable to misinformation or disinformation, as patients seek to gather as much information as they can during this time [18].

Medical societies, including the EAU, are working address these critical moments and provide digestible information for patients and their loved ones. The foundation for all diagnostic and therapeutic decisions is the set of guidelines. These guidelines are directed at physicians and incorporate all available evidence according to specific criteria. The evidence is synthesized by guideline panels to enhance understandability and provide a comprehensive overview for the general physician [27]. Subsequently, the EAU patient office aims to make this information available for patients [6]. As described, this process is time-consuming and resource intensive. Our study demonstrates a technological alternative: GenAI can generate PEMs from guideline information within one minute at minimal or no costs.

Neither original PEMs nor GPT-4 generated PEMs are flawless, despite an overall agreement on correctness of the information by the expert raters in our study. While not revealing statistically significant differences in overall correctness,

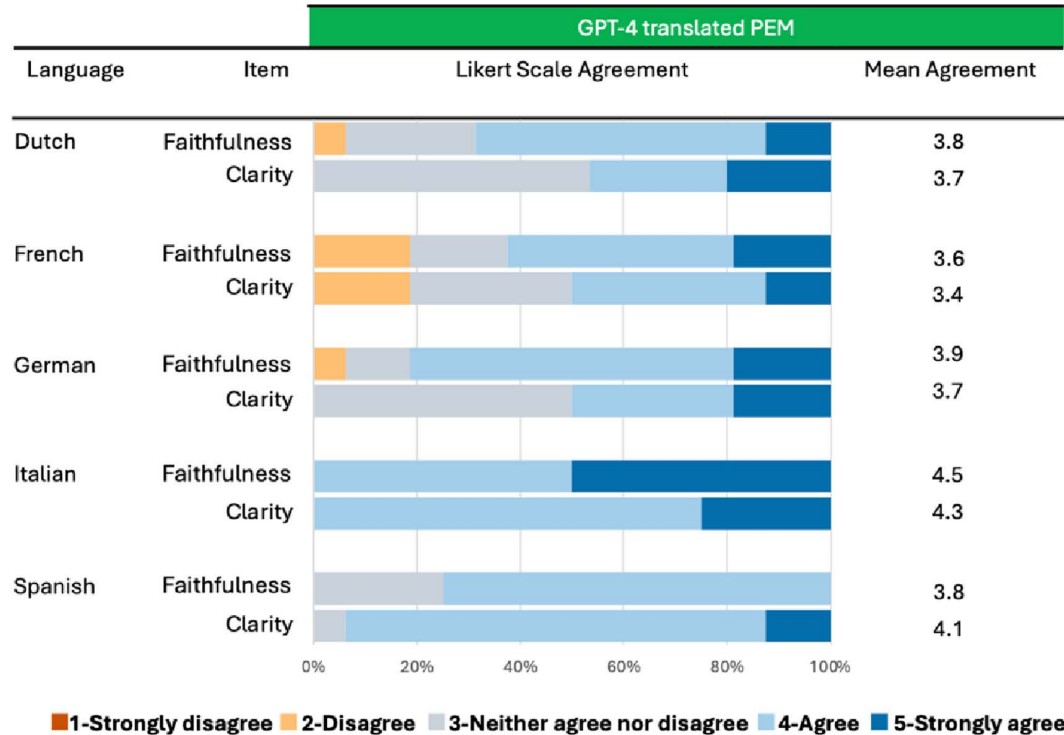

**Fig 4. Translation of patient education material (PEM).** Faithfulness and clarity of GenAI generated PEMs for further dissemination, across 5 commonly spoken EU languages. Abbr.: Patient educational material (PEM). European Union **(EU).**

we observe the tendency towards more raters that disfavor GPT-4 generated PEMs for localized disease and original PEMs for metastatic disease. The drawbacks and limitations of original PEMs and GPT-4 generated PEMs might arise from different sources. Original PEMs might not be updated regularly. For instance, the information on bladder cancer for metastatic disease does not cover novel developments regarding immunotherapy and combination therapies with antibody drug conjugates such as enfortumab vedotin, despite entering routine care [28,29]. This may contribute to the favoring of GenAI-generated PEMs which can be more easily updated in tandem with guideline releases. A clear limitation of the workflow for original PEMs development might be expressed in the availability of the information across different languages. On the website, PEMs was provided for only two out of six selected languages for all cancer entities analyzed in this study. This is in contradiction to the efforts of different bodies of the EU where it is common practice to translate medical information into the official languages of all member states [30]. Conversely, GPT-4 generated PEMs might be limited in identifying all relevant information from the provided input or providing information in the wrong context [31]. While GenAI-generated PEMs showed improved readability, they achieved similar levels of correctness and quality in representing the guidelines as the original PEMs. Given that readability metrics do not consider the quality of the writing, optimizing for readability is not sufficient for ensuring proper dissemination of PEMs [32]. Resolving the limitations may be possible through integration of GenAI into the workflow of updating PEMs. Integration with human verification and editing might allow organizations to keep up with the workload of guideline changes and increase readability while also maintaining the quality (Fig 1). This same workflow extends to translating PEMs. While the language capabilities will likely improve with future models [33], native-speaking urologists and certified translators should always work to finalize the translations prior to widespread dissemination.

When communicating healthcare information, adjusting to the literacy and knowledge level of the target audience is essential [34]. GenAI may represent a leap forward in providing readable and tailored PEMs. We propose that PEMs could be adjusted to the education level of patients, their language, and the level of empathy and knowledge that a patient requires. Our study reveals that GenAI can provide information at a predefined readability level and generate the output in various languages. Empathy and knowledge are two future aspects that will have to be addressed in PEM generation but are features that GenAI can solve in theory [35].

Previous studies have asked patient questions directly to GPT-4 and relied on the proprietary knowledge domain [36,37]. In these studies, authors found that results were mostly correct with several obvious mistakes made by the GenAI, as models like GPT-4 are trained on the entire internet [38]. The current approach of using a guideline as a validated knowledge source is a step forward and potentially limits the error rate. However, human oversight is still required in this approach to ensure ethical use and responsible dissemination for upholding the integrity of the output [39,40].

Regarding ethical practice, initiatives are underway to ensure the ethical and responsible use of GenAI as an intervention in academic research [40]—an essential step for piloting patient-facing tools. Similar methods have been spearheaded by the EU through the AI Act of 2024, which classifies patient-facing GenAI models as "low risk" and requires proper disclosure with oversight by individuals with sufficient "AI literacy" [41].

GenAI will have tremendous impact on the patient-physician relationship. As for the generated PEMs, patients can potentially detect that they were written by AI models, and it remains largely unknown how patients will react to material that they recognize as AI generated. However, patients reveal high levels of trust in physicians incorporating AI into the diagnostic or therapeutic pathway [42]. Additionally, patients are surrounded by easily digestible yet inaccurate medical information through social media and the internet which may be preferred to society endorsed PEMs [3,17]. Therefore, an integration of AI with human editing and endorsement is the logical next step to enhance the dissemination and adoption of validated information about disease and treatment.

Several limitations must be considered. First, the number of expert reviewers in the randomized assessment was limited due to the availability of YAU working group members. While steps were taken to minimize the potential bias of reviewers using a single-blinded, randomized assessment, bias can still be present given the distinct syntactic and lexical hallmarks found with GenAI output [43,44]. Further, while Likert scales are often used in the evaluation of GenAI-generated content [11,13], the ability to detect differences is often limited given the clustering of responses. Validated tools for evaluating the quality of GenAI-generated content are needed. Moreover, GenAI technology evolves quickly - while initially guidelines were input as text chunks, they can now be input as entire websites of PDF files directly and therefore integrated seamlessly into GPTs. Given this, the reproducibility of these findings is difficult to predict over longer timespans, however GenAI tools should regularly be reassessed to insure similar or improved performance. Finally, the understandability of the content has not been tested in patients. However, readability scores serve as an established tool with defined thresholds [5]. Additionally, the Bridging Readable and Informative Dissemination with GenerativE Artificial Intelligence (BRIDGE-AI) Initiative (OSF: https://osf.io/8yz6d/) is actively working with patient advocacy groups representing major genitourinary cancers to conduct the first randomized controlled trial evaluating how well urology patients understand GenAI-generated content.

## Conclusion

GenAI can be used to extract information from urological guidelines and synthesize the information at a better readability level and with the same correctness as information that is provided by PEMs released by the EAU patient office. Leveraging the information extraction and language capabilities of GenAI and integrating them into a workflow with human oversight can tremendously lower the workload to ensure up-to-date and correct PEMs. Early embrace of this new technology by medical societies may improve PEMs dissemination, especially in multilingual regions such as the EU.

## Supporting information

**S1 Fig. Pilot study for correctness assessment of English patient education materials (PEM).** Original patient educational materials (PEM) and GPT-4 generated PEM were used for analysis. Information for both localized and metastatic disease were evaluated for accuracy, completeness, and clarity on a 5-item likert scale. Evaluation was performed in a single-blinded, randomized manner. Abbr.: Patient educational material (PEM).
(TIF)

**S1 Table. Checklist for Reporting Results of Internet E-Surveys (CHERRIES).** This checklist has been modified from Eysenbach G. *Improving the quality of Web surveys: the Checklist for Reporting Results of Internet E-Surveys (CHERRIES).* J Med Internet Res. 2004 Sep 29;6(3):e34 [erratum in J Med Internet Res. 2012;14(1):e8]. Article available at https://www.jmir.org/2004/3/e34/; erratum available https://www.jmir.org/2012/1/e8/. Copyright ©Gunther Eysenbach. Originally published in the *Journal of Medical Internet Research*, 29.9.2004 and 04.01.2012. This is an open-access article distributed under the terms of the Creative Commons Attribution License (https://creativecommons.org/licenses/by/2.0/), which permits unrestricted use, distribution, and reproduction in any medium, provided the original work is properly cited.
(DOCX)

**S2 Table. Prompt for PEM language translation.** Used to translate the GenAI-generated PEMs into the respective language within the EU. Abbr.: Patient educational material (PEM). European Union (EU).
(DOCX)

**S3 Table. Availability of original PEM.** Availability of PEMs in spoken EU languages for prostate, kidney, bladder, and testicular cancer EAU treatment guidelines, based on October 10, 2023 query of the EAU patient portal (https://patients.uroweb.org). Abbr.: Patient educational material (PEM). European Association of Urology (EAU).
(DOCX)

**S4 Table. Baseline characteristics of raters assessing correctness of PEM.** Characteristics of study participants displayed based on YAU working group and language background. All native speakers are fluent in English and one of the five investigated EU-spoken languages. Abbr.: Young Academic Urologists (YAU).
(DOCX)

**S1 Raw Data. Raw Data for PEM evaluation.** Data file with raw data used for statistical analysis comparing patient educaiton matierals (PEMs) created by GAI vs Humans.
(XLSX)

## Acknowledgments

### YAU Collaborative authorship

Ignacio Puche-Sanz[1], Inês Anselmo da Costa Santiago[2], Josias Bastian Grogg[3], Jorge Caño-Velasco[4], Luca Antonelli[5,6], Martina Maggi [7], Francesco Del Giudice [7], Michele Marchioni [8], Laura Ibañez Vazquez[9], Zhenjie Wu [10], Riccardo Bertolo[11], João Lobo [12], Pia Pfaffenholz[13,14], Vittorio Fasulo[15,16], Daniele Amparore[17], Luca Afferi[18], Elisabeth Grobet-Jeandin [19], Laura Marandino[20], Renate Pichler[21], Ugo Giovanni Falagario[22], Gaelle Margue[23]

[1] Department of Urology, Hospital Universitario Virgen de las Nieves, Granada, Spain
[2] Klinikum am Urban, Klinik für Urologie, Berlin, Germany
[3] University Hospital Zurich, University of Zurich, Zurich, Switzerland
[4] Department of Urology, Hospital General Universitario Gregorio Marañon, Madrid, Spain
[5] Department of Urology, Luzerner Kantonsspital, University of Lucerne, Switzerland
[6] Department of Urology, Sapienza University of Rome, Rome, Italy

[7] Department of Maternal Infant and Urologic Sciences, "Sapienza" University of Rome, Policlinico Umberto I, Rome, Italy

[8] Department Of Medical Oral and Biotechnological Sciences. "G. d'Annunzio" University of Chieti, Chieti, Italy

[9] Department of Urology, Health Research Institute, Hospital Clinico San Carlos, Madrid, Spain

[10] Department of Urology, Changhai Hospital, Naval Medical University, Shanghai, China

[11] Urology Unit, Azienda Ospedaliera Universitaria Integrata (AOUI) Verona, Borgo Trento Hospital, University of Verona, Verona, Italy

[12] Portuguese Oncology Institute of Porto (IPO Porto)/Porto Comprehensive Cancer Center Raquel Seruca (Porto.CCC Raquel Seruca), 4200–072 Porto, Portugal

[13] Department of Urology, Uro-Oncology, Robot Assisted and Reconstructive Urologic Surgery, University of Cologne Faculty of Medicine and University Hospital Cologne, Cologne Germany

[14] Center for Integrated Oncology (CIO) Köln-Bonn, Cologne, Germany

[15] Department of Biomedical Sciences, Humanitas University, Pieve Emanuele, MI, Italy

[16] Department of Urology, IRCCS-Humanitas Research Hospital, Rozzano, MI, Italy

[17] Department of Oncology, Division of Urology, University of Turin, San Luigi Gonzaga Hospital, Orbassano, Italy

[18] Department of Urology, Luzerner Kantonsspital, Luzern, Switzerland

[19] Urology Department, Hôpitaux Universitaires de Genève, Rue Gabrielle-Perret-Gentil 4, 1205, Geneva, Switzerland

[20] The Royal Marsden Hospital NHS Foundation Trust, London, UK

[21] Department of Urology, Medical University of Innsbruck, Innsbruck, Austria

[22] Department of Urology, University of Foggia, Foggia, Italy; Department of Molecular Medicine and Surgery, Karolinska Institutet, Stockholm, Sweden

[23] Department of Urology, Bordeaux University Hospital, Bordeaux, France

## Author contributions

**Conceptualization:** Severin Rodler, Serena Maruccia, Giovanni E. Cacciamani.

**Data curation:** Severin Rodler, Francesco Cei, Conner Ganjavi, Giovanni E. Cacciamani.

**Formal analysis:** Severin Rodler, Giovanni E. Cacciamani.

**Investigation:** Severin Rodler.

**Methodology:** Severin Rodler, Giovanni E. Cacciamani.

**Supervision:** Conner Ganjavi, Enrico Checcucci, Ines Rivero Belenchon, Stefano Puliatti, Pietro Piazza, Loïc Baekelandt, Karl-Friedrich Kowalewski, Christian D. Fankhauser, Marco Moschini, Giorgio Gandaglia, Riccardo Campi, Andre De Castro Abreu, Giorgio I. Russo, Andrea Cocci, Serena Maruccia, Giovanni E. Cacciamani.

**Visualization:** Conner Ganjavi.

**Writing – original draft:** Severin Rodler, Francesco Cei.

**Writing – review & editing:** Severin Rodler, Francesco Cei, Conner Ganjavi, Enrico Checcucci, Pieter De Backer, Mark Taratkin, Alessandro Veccia, Juan Gómez Rivas, Giovanni E. Cacciamani.

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
