## [Decision Letter · Decision Letter 0]

4 Mar 2025

PONE-D-24-54284GPT-4 GENERATES ACCURATE AND READIBLE PATIENT EDUCATION MATERIALS ALIGNED WITH CURRENT ONCOLOGICAL GUIDELINES. A RANDOMIZED ASSESSMENT

Dear Dr. Cacciamani,

Thank you for submitting your manuscript "Guideline-based patient educational materials using generative artificial intelligence" to PLOS ONE. Please accept our sincere apologies for the delay in getting this report to you. The difficulty in securing appropriate reviewers with expertise in both urological education materials and artificial intelligence applications resulted in a longer review process than anticipated.

We have now received comments from one reviewer, and I have also personally reviewed the manuscript in detail - See this under editor comments below. Both reviews identify several strengths in your work, particularly the novel application of GenAI to creating patient educational materials with improved readability while maintaining comparable accuracy to traditional methods.

However, there are several methodological issues and areas for improvement that need to be addressed before your manuscript can be considered further. I have attached both the reviewer's comments and my own editorial assessment for your consideration.

Key areas requiring attention include:

Methodological details regarding the GenAI framework and prompting approach

Justification for sample sizes and assessment methodology

Statistical reporting and analysis of the translation quality findings

Enhanced discussion of limitations and regulatory considerations

Please provide a point-by-point response to all comments, either implementing the suggested changes or providing a clear rationale for why certain suggestions cannot be accommodated. 

If you require any clarification regarding the comments or need assistance with the revision process, please do not hesitate to contact our editorial office Please submit your revised manuscript by Apr 18 2025 11:59PM. If you will need more time than this to complete your revisions, please reply to this message or contact the journal office at plosone@plos.org . Please include the following items when submitting your revised manuscript:

We look forward to receiving your revised manuscript.

Kind regards,

Isaac Amankwaa, Ph.D.

Guest Editor

PLOS ONE

“NONE”

4. In the online submission form you indicate that your data is not available for proprietary reasons and have provided a contact point for accessing this data. Please note that your current contact point is a co-author on this manuscript. According to our Data Policy, the contact point must not be an author on the manuscript and must be an institutional contact, ideally not an individual. Please revise your data statement to a non-author institutional point of contact, such as a data access or ethics committee, and send this to us via return email. Please also include contact information for the third party organization, and please include the full citation of where the data can be found.

**Additional Editor Comments:**

AbstractLine 184, page 4, under abstract – the patient summary could be slightly expanded to match the level of detail in the rest of the abstract while maintaining its improved readability. Also, ‘patient info,’ should be written in full.Line 161, page 4 – Young academic urologist (YAU).. revise to ensure consistency in capitalisation.IntroductionConsider starting with the challenge of health literacy and readable PEMs before mentioning internet statisticsA brief mention of previous research, specifically on AI in medical education materials, would strengthen your argumentAdd a sentence addressing the potential ethical implications or limitations of GenAI in medical informationConsider including a line about the novelty/gap your study addressesLine 206: "on yearly bases" should be "on a yearly basis"Line 214: "different scholarity levels" - Consider using "educational" or "literacy" levels instead of "scholarity"Line 211: "but lacks consistent correctness when relying on the proprietary knowledge domain" - This phrase is somewhat unclear; consider rewording for clarityLine 209: Consider rephrasing "not meeting requirements set by the European Union (EU) for understandability of clinical summaries" for better clarityMethodsThe "GPT-4 framework development and custom-GPT creation" mentioned in the opening paragraph is never fully explained. Please provide some explanation to complement the figure provided to ensure reproducibility.Regarding your prompting methodology for GPT-4, while you mention that "prompts were based on previously published methods and modified to extract guideline information," this provides insufficient detail for reproducibility. Please consider enhancing this section by:Including at least one complete example promptSpecifying the exact modifications made to previously published methodsDetailing any GPT-4 parameters used (temperature, token limits, etc.)Explaining how your prompts specifically targeted EU-recommended reading levelsDescribing any iterative refinement process used to optimize the prompt.Regarding the sample size of reviewers (12 for pilot, 20 for main assessment with 5 per cancer type), I note the absence of a formal justification for these numbers. While traditional power calculations may be challenging for expert review studies, I recommend:Providing a brief rationale for the chosen sample sizes based on: Previous similar validation studies in medical education/PEM literaturePractical considerations regarding YAU expert availabilityAnticipated effect sizes based on pilot resultsAcknowledging this limitation and discussing how you addressed potential reliability concerns with a relatively small number of reviewers per cancer typeThere is no mention of translation validation methodology or back-translation verification.No explanation of how specific PEMs were selected for evaluation or whether they represent the full spectrum of urological cancers.

**Results **

Line 356: Mentions "translations to Dutch, French, German, Italian and French" - French is listed twice, likely a typoLine 386: p-value written as "p=738." instead of "p=.738"Line 372-373: Only reports median values (4) for all measures in the pilot study, making it impossible to detect nuanced differencesThe study reports numerous non-significant p-values but no effect sizes or confidence intervalsNo interrater reliability metrics (e.g., Fleiss' kappa) are reported despite multiple ratersResults show identical median values (mostly 4) for almost all comparisons, raising questions about the sensitivity of your measurement scale

Discussion

No discussion of why the readability metrics were better for GenAI but accuracy and clarity ratings were similarLimited exploration of the variability in translation quality (simply stating "majority of cases" is imprecise)No explanation for the seemingly contradictory finding that GenAI PEMs were less favored for localized disease but more favored for metastatic disease (line 421-423)Limited connection to broader AI ethics literature or frameworks for medical AI implementationNo discussion of regulatory considerations for AI-generated patient materialsMinimal connection to patient communication theory or health literacy researchNeed to include a limitation section that:discusses the potential limitations in the assessment methodology (small sample of expert reviewers, potential biases)discusses Likert scale's ability to detect meaningful differences when most ratings clustered around 4reflects on whether urologists are the ideal evaluators of patient-oriented materialsany other limitations

Reviewers' comments:

Reviewer's Responses to Questions

**Comments to the Author**

1. Is the manuscript technically sound, and do the data support the conclusions?

Reviewer #1: Yes

2. Has the statistical analysis been performed appropriately and rigorously? 

Reviewer #1: Yes

3. Have the authors made all data underlying the findings in their manuscript fully available?

Reviewer #1: Yes

4. Is the manuscript presented in an intelligible fashion and written in standard English?

Reviewer #1: Yes

5. Review Comments to the Author

Reviewer #1: Abstract, Line 158:

The sentences "A randomized assessment of the correctness of these materials was conducted by 32 urologists and urology residents." and "Thirty-two members of the Young academic urologists (YAU) groups evaluated the accuracy, completeness, and clarity of the original versus GPT-generated PEMs." seem to repeat the same information. Please use just one of them to avoid redundnacy.

This same comment applies to the same section for the sentences "Additionally, translations into other languages were evaluated for correctness and clarity by native-speaker urologists." and "The translation assessment involved two native speakers from different YAU groups for each language: Dutch, French, German, Italian, and Spanish."

Abstract, results: Please mention the number of original PEMs and AI generated PEMs assessed here, not just in the main results section.

The fact that translations of GPT-generated PEMs were rated as correct in 77.5% of cases and clear in 67.5% of cases, which indicates an error rate that should not be neglected and that AI is still not very reliable to translate such information without human supervision to avoid misinformation and chaos. Please elaborate on this in the discussion and highlight as a point of weakness in these generated PEMs, aside from highlighting their strength in more proper readability when compared to original PEM.

FKRE should stand for Flesch Kincaid Reading Ease, not Flesch Reading Ease. Please correct it in the entire manuscript.

Results, Line 368: The readability assessment is represented in figure 2, and not figure 1 as the text claims. Please correct it. Also, in line 388. The correctness assessment is represented in figure 3, and not figure 2 as the text says. The same error is i line 395 (Should be figure 4 instead of 3).

There is no Supplementary Table 3, and there is a jump from Supp. Table 2 to 4. Please rename 4 to 3 and correct in the text as well.

6. PLOS authors have the option to publish the peer review history of their article (what does this mean? ). If published, this will include your full peer review and any attached files.

**Do you want your identity to be public for this peer review?** For information about this choice, including consent withdrawal, please see our Privacy Policy .

Reviewer #1: No

---

## [Author Response · Author response to Decision Letter 1]

30 Mar 2025

Journal Requirements And Editor Comments

Thank you for submitting your manuscript "Guideline-based patient educational materials using generative artificial intelligence" to PLOS ONE. Please accept our sincere apologies for the delay in getting this report to you. The difficulty in securing appropriate reviewers with expertise in both urological education materials and artificial intelligence applications resulted in a longer review process than anticipated.

We have now received comments from one reviewer, and I have also personally reviewed the manuscript in detail - See this under editor comments below. Both reviews identify several strengths in your work, particularly the novel application of GenAI to creating patient educational materials with improved readability while maintaining comparable accuracy to traditional methods.

However, there are several methodological issues and areas for improvement that need to be addressed before your manuscript can be considered further. I have attached both the reviewer's comments and my own editorial assessment for your consideration.

Response: We thank the editor for the diligence in finding sufficient reviewers for this manuscript given the expertise of the reviewers required, and we understand the delay as part of the publishing process. We are deeply grateful to the editor for providing a review of the manuscript, ensuring it undergoes a thorough review and is made more robust in exploring the role of GenAI in patient education materials (PEM). We appreciate the reviewers’ comments regarding the novel utility of GenAI in this space and will address the methodological and other concerns in the responses below.

Response: We have modified the manuscript to accurately reflect the PLOS ONE style requirements, ensuring proper formatting and file naming.

Response: No code was used in this study.

“NONE”

Response: We have modified the cover letter to properly include recently developed conflicts of interest and will include the statement in the online submission.

4. In the online submission form you indicate that your data is not available for proprietary reasons and have provided a contact point for accessing this data. Please note that your current contact point is a co-author on this manuscript. According to our Data Policy, the contact point must not be an author on the manuscript and must be an institutional contact, ideally not an individual. Please revise your data statement to a non-author institutional point of contact, such as a data access or ethics committee, and send this to us via return email. Please also include contact information for the third party organization, and please include the full citation of where the data can be found.

Response: We thank the editor for the comment. Please consider that the initial page of the manuscript was included for internal purposes and is not applicable for the submission of the manuscript. We have eliminated this page as it does not comply with the PLOS guidelines for submission.

Response: Captions have been appropriately included for all main figures/tables in the manuscript.

Response: Captions have been appropriately added for all supplemental figures/tables.

Response: Citations have been reviewed to ensure accuracy and completeness.

Additional Editor Comments

Abstract:

1) Line 184, page 4, under abstract – the patient summary could be slightly expanded to match the level of detail in the rest of the abstract while maintaining its improved readability. Also, ‘patient info,’ should be written in full.

Response: We thank the editor for the comment and have expanded the patient summary as follows:

“Some cancer facts made for sick people can be hard to read or not in the right words for those with prostate, bladder, kidney, or testicular cancer. This study used AI to quickly make short and easy-to-read content from trusted facts. Doctors checked the AI content and found that they were just as accurate, complete, and clear as the original text made for patients. They also worked well in many languages. This AI tool can assist providers in making it easier for patients to understand their cancer and the best care they can get.” (Page 3, lines 129-135)

2) Line 161, page 4 – Young academic urologist (YAU).. revise to ensure consistency in capitalisation.

Response: We thank the editor for the correction and have revised YAU as “Young Academic Urologists” across all instances.

Introduction:

1) Consider starting with the challenge of health literacy and readable PEMs before mentioning internet statistics.

Response: We thank the editor for the comment and agree that health literacy is a major challenge and should be featured at the start of the introduction. We have modified the initial paragraph to include the sentences as follows:

“Health literacy remains an often-overlooked factor in health disparities, with research directly linking low literacy to negative health outcomes[1, 2]. This is in part due to the dearth of validated medical information that is written specifically for patients, forcing many of them to turn to less reliable sources on the internet and potentially compromise care[3]. Approximately 4.5% of all internet search queries are health-related, totaling around 6.75 million health-related searches each day[4]. Given this, countering the exposure to potential misinformation through the creation of readable, correct, and up-to-date patient educational materials (PEMs) is a fundamental objective of societies across specialties[5].” (Page 4, lines 147-154)

Studies cited as follows:

N. D. Berkman, S. L. Sheridan, K. E. Donahue, D. J. Halpern, and K. Crotty, "Low health literacy and health outcomes: an updated systematic review," (in eng), Ann Intern Med, vol. 155, no. 2, pp. 97-107, Jul 19 2011, doi: 10.7326/0003-4819-155-2-201107190-00005.

[G. Bartlett, R. Blais, R. Tamblyn, R. J. Clermont, and B. MacGibbon, "Impact of patient communication problems on the risk of preventable adverse events in acute care settings," (in eng), Cmaj, vol. 178, no. 12, pp. 1555-62, Jun 3 2008, doi: 10.1503/cmaj.070690.

S. Loeb et al., "Fake News: Spread of Misinformation about Urological Conditions on Social Media," (in eng), Eur Urol Focus, vol. 6, no. 3, pp. 437-439, May 15 2020, doi: 10.1016/j.euf.2019.11.011.

G. Eysenbach and C. Kohler, "What is the prevalence of health-related searches on the World Wide Web? Qualitative and quantitative analysis of search engine queries on the internet," (in eng), AMIA Annu Symp Proc, vol. 2003, pp. 225-9, 2003.

B. Weis, "Health Literacy: A Manual for Clinicians. Chicago, IL: American Medical Association, American Medical Foundation; 2003.• National Institutes of Health. How to Write Easy to Read Health Materials: National Library of Medicine Website," How to Write Easy to Read Health Materials: National Library of Medicine Website.

2) A brief mention of previous research, specifically on AI in medical education materials, would strengthen your argument.

Response: We agree with the editor’s comment and have included a sentence with two cited studies, one in ENT and one in urology, which have used generative-AI tools for the specific goal of simplifying patient education materials. This demonstrates the potential for improved PEMs using GenAI as discussed in the literature.

The sentence added is as follows:

“Previous studies have shown that GenAI tools like ChatGPT can be used to simplify patient education materials in multiple specialties with comparable correctness and improved readability.” (Page 4, lines 172-174)

Citations are as follows:

Y. B. Shah, A. Ghosh, A. Hochberg, J. R. Mark, C. D. Lallas, and M. S. Shah, "Artificial intelligence improves urologic oncology patient education and counseling," (in eng), Can J Urol, vol. 31, no. 5, pp. 12013-12018, Oct 2024.

E. A. Patel et al., "The Use of Artificial Intelligence to Improve Readability of Otolaryngology Patient Education Materials," (in eng), Otolaryngol Head Neck Surg, vol. 171, no. 2, pp. 603-608, Aug 2024, doi: 10.1002/ohn.816.

3) Add a sentence addressing the potential ethical implications or limitations of GenAI in medical information.

Response: We thank the editor for the comment regarding the important discussion of ethical implications and limitations with this technology. Regarding the limitations, these are briefly discussed in the introduction citing the study by Davis et al. which identified limitations in “consistent correctness when relying on the proprietary knowledge domain”. We feel this inclusion is a sufficient introduction to the limitations, however we will expand on the limitations of GenAI in the discussion section. Regarding the ethical implications, we have included a reference to Eppler et al., which conducted a global survey of urologists on the use of GenAI. The study found that while urologists are incorporating GenAI in patient education, they remain concerned about ethical issues including the risk of bias and patient privacy. We have added a sentence exploring these ethical issues as follows:

“However, a global survey of urologists found that while GenAI is useful for patient education, there are ethical concerns regarding the potential spread of biased information and patient privacy breaches[12].” (Page 4, lines 167-169)

Citation is as follows:

M. Eppler et al., "Awareness and Use of ChatGPT and Large Language Models: A Prospective Cross-sectional Global Survey in Urology," (in eng), Eur Urol, vol. 85, no. 2, pp. 146-153, Feb 2024, doi: 10.1016/j.eururo.2023.10.014.

R. Davis et al., "Evaluating the Effectiveness of Artificial Intelligence-powered Large Language Models Application in Disseminating Appropriate and Readable Health Information in Urology," (in eng), J Urol, vol. 210, no. 4, pp. 688-694, Oct 2023, doi: 10.1097/ju.0000000000003615.

4) Consider including a line about the novelty/gap your study addresses

Response: We thank the reviewer for the comment and have added a sentence elucidating the novelty of the study, which is primarily in the use of guidelines as the data source for extraction using custom GenAI model to improve the accuracy of PEMs. Another novelty is the translation of the PEMs into various EU languages to fill the gaps in current PEMs which are written only in English and German for all four urologic cancers. The sentences added and modified are as follows:

“In this study we assessed the potential of GenAI to extract guideline information from the European Association of Urology (EAU), simplify it for better patient understanding, and translate it into multiple languages - bridging the accessibility gap for certified healthcare content." (Page 4, lines 170-172)

“While these studies relied on the base GenAI model, our study represents a novel approach through the creation of a custom model to extract information from recent urologic oncology guidelines, mimicking the standard workflow for PEM development.” (Page 4, lines 174-177)

5) Line 206: "on yearly bases" should be "on a yearly basis"

Line 214: "different scholarity levels" - Consider using "educational" or "literacy" levels instead of "scholarity"

Line 211: "but lacks consistent correctness when relying on the proprietary knowledge domain" - This phrase is somewhat unclear; consider rewording for clarity

Line 209: Consider rephrasing "not meeting requirements set by the European Union (EU) for understandability of clinical summaries" for better clarity

Response: We thank the editor for their suggested improvements and have incorporated them into the introduction for increased clarity.

Methods:

1) The "GPT-4 framework development and custom-GPT creation" mentioned in the opening paragraph is never fully explained. Please provide some explanation to complement the figure provided to ensure reproducibility.

Response: We thank the editor for the comment. In the section on “GenAI Framework Development”, we discuss the development of the tri-phasic framework which builds off the prompt in Eppler et al. and can be found in the supplementary materials of that study. The framework in Eppler et al. was designed as a 2-layer system to extract relevant information from text and summarize it at a particular reading level. The tri-phasic framework as outlined in our methods “was utilized to extract relevant information from clinical guidelines (phase 1) and adapt it to the readability level recommended by the European Union (phase 2), as well as to translate the English output into languages commonly spoken within the EU (Phase 3).” This framework was distilled into a prompt and used to build a custom GPT that has been linked in the methods section https://chatgpt.com/g/g-sZZBNfGSl-patient-education-material-readible-summaries. The translation prompt has been included the supplementary materials (S2 Table). To remove any confusion and discrepancy with Figure 1, the key portions of the triphasic framework are now highlighted in the Figure 1 caption as follows:

“Figure 1. Pipeline for generating layperson summaries of medical trials using a tri-phasic large language model (LLM)-based framework with human oversight. The process starts with input from the most current guidelines, which the LLM processes in stages. First, key sections (localized disease, metastatic disease and treatments of each section) are identified (phase 1) and simplified into lay terms (phase 2). The output is then translated into five languages commonly spoken in the EU (phase 3).”

2) Regarding your prompting methodology for GPT-4, while you mention that "prompts were based on previously published methods and modified to extract guideline information," this provides insufficient detail for reproducibility. Please consider enhancing this section by:

• Including at least one complete example prompt

• Specifying the exact modifications made to previously published methods

• Detailing any GPT-4 parameters used (temperature, token limits, etc.)

• Explaining how your prompts specifically targeted EU-recommended reading levels

• Describing any iterative refinement process used to optimize the prompt.

Response: We thank the editor for their comment regarding prompting methodology and agree that clarification of the

---

## [Editor Report · Decision Letter 1]

22 Apr 2025

GPT-4 GENERATES ACCURATE AND READABLE PATIENT EDUCATION MATERIALS ALIGNED WITH CURRENT ONCOLOGICAL GUIDELINES. A RANDOMIZED ASSESSMENT

PONE-D-24-54284R1

Dear Authors,

We’re pleased to inform you that your manuscript has been judged scientifically suitable for publication and will be formally accepted for publication once it meets all outstanding technical requirements.

Kind regards,

Isaac Amankwaa, Ph.D.

Guest Editor

PLOS ONE
---

## [Editor Report · Acceptance letter]

PONE-D-24-54284R1

PLOS ONE

Dear Dr. Cacciamani,

I'm pleased to inform you that your manuscript has been deemed suitable for publication in PLOS ONE. Congratulations! Your manuscript is now being handed over to our production team.

Kind regards,

on behalf of

Dr. Isaac Amankwaa

Guest Editor

PLOS ONE